# Efficient Hardware Scaling and Diminishing Returns in Large-Scale Training of Language Models

**Jared Fernandez**                                                    *jaredfern@cmu.edu*
*Carnegie Mellon University*

**Luca Wehrstedt**                                                     *lcw@meta.com*
*Meta FAIR*

**Leonid Shamis**                                                      *lshamis@meta.com*
*Meta FAIR*

**Mostafa Elhoushi**                                                   *melhoushi@meta.com*
*Meta FAIR*

**Kalyan Saladi**                                                      *skalyan@meta.com*
*Meta FAIR*

**Yonatan Bisk**                                                       *ybisk@cs.cmu.edu*
*Carnegie Mellon University*

**Emma Strubell**                                                      *strubell@cmu.edu*
*Carnegie Mellon University*

**Jacob Kahn**                                                         *jacobkahn@meta.com*
*Meta FAIR*

**Reviewed on OpenReview:** `https://openreview.net/forum?id=p7jQEf3wlh`

## Abstract

To train the exceedingly large neural networks required in modern applications, such as large language models (LLMs), model training is distributed across tens of thousands of hardware accelerators (e.g. GPUs), requiring orchestration of computation and communication across large computing clusters. In this work, we demonstrate that careful consideration of hardware configuration and parallelization strategy is critical for effective (i.e. compute- and cost-efficient) scaling of model training. We conduct an extensive empirical study of the performance of large-scale LLM training workloads across model size, hardware configurations, and distributed parallelization strategies with current best practices. In experiments with model sizes up to 70B parameters and utilizing up to 2048 H100 GPUs, we demonstrate that: (1) Naive scale out with Fully Sharded Data Parallelism (FSDP) incurs communication overhead which leads parallelization strategies previously thought to be sub-optimal to in fact become preferable; and (2) scaling the total number of accelerators for training quickly yields diminishing returns even when hardware and parallelization strategies are properly optimized, implying poor marginal performance per additional unit of power or GPU-hour.

## 1 Introduction

The increasing size of state-of-the-art neural language models, which now contain in excess of hundreds of billions of parameters, yields larger computational workloads and memory requirements during training. In this regime, the memory requirements from increasing numbers of model parameters and large batch sizes are such that the model parameters, activations, and optimizer states required for model training no longer fit within the memory of a single

GPU accelerator. To address the memory limitations of a single device and to leverage the increased processing power of additional accelerators, the largest workloads necessitate distribution across thousands of hardware accelerators (i.e. GPUs and TPUs).

The need for training algorithms for distributing workloads across large numbers of accelerators has motivated the development of various data and model parallelism strategies – discussed in more detail in §2 (Rasley et al., 2020; Shoeybi et al., 2019; Zhao et al., 2023; Li et al., 2020; Ryabinin et al., 2023). Combining data, tensor and pipeline parallelism (3D parallelism) and sharded data parallelism (FSDP and DeepSpeed ZeRO) have been developed as primary methods to address memory limitations during training (Shoeybi et al., 2019; Shazeer et al., 2017; Lepikhin et al., 2020). In particular, sharded data parallelism without model parallelism has emerged as one of the most common methods for langauge model training and been used in the training of open models such as: OLMo (Groeneveld et al., 2024), IBM Granite (Granite Team, 2024), Apple OpenELM (Mehta et al., 2024), and Mosaic MPT (Team, 2023).

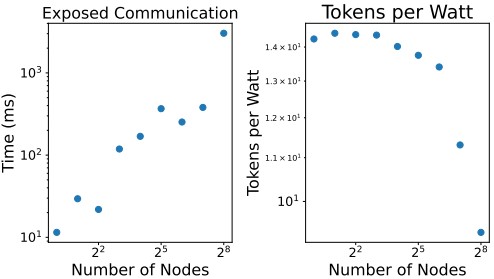

Figure 1: Despite minimal communication overhead on less than 32 nodes, increasing communication overhead leads FSDP to observe diminishing returns on power efficiency with over 30% reduction at scale.

Although many theoretical cost models have been developed to estimate the communication and computation performance of various parallelization methods (Qi et al., 2017; Cai et al., 2021; Pal et al., 2019; Gholami et al., 2018; Jia et al., 2019), existing approaches do not account for the full variety of components in modern training systems, including: model architecture, network topology, parallelisms, hardware speeds and architectures. Previous work has empirically studied the performance and scaling properties of 3D parallelism (Narayanan et al., 2021; Hagemann et al., 2023), the scaling and efficiency properties of *sharded parallelism strategies* and their interactions with model parallelism techniques are less well studied; despite its prevalence in practice (e.g. OLMo, Granite, OpenELM, MPT).

While there are stable distributed training recipes that perform well at large scale, the procedure for deciding on such configurations and their scaling properties are not well understood or documented; and the regimes in which selected parallelism strategies are communication and computation efficient is often unspecified. Previous work (Narayanan et al., 2021; Hagemann et al., 2023) has studied the effects of various forms of model parallelism on training efficiency; we expand on this direction with studies across larger ranges of hardware and investigations of parallelism configurations not covered in previous studies. In particular, we consider the effects of Fully Sharded Data Parallelism (FSDP) on training efficiency and observe that its integration substantially impacts the choice of optimal training configurations. We show that prior work and existing best practices determined with model parallelism in the absence of FSDP, yield suboptimal performance and efficiency when combined with sharded data parallelism strategies. In addition, we conduct measurements of GPU power utilization and demonstrate that these existing approaches yield dramatically worse power efficiency, potentially worsening the energy and environmental cost of machine learning research and development (Strubell et al., 2019; Luccioni et al., 2024; Schwartz et al., 2020)

In this work, we conduct an extensive empirical study across both parallelization strategies and hardware scales; and we contribute the following:

- A **large-scale empirical study** of distributed training across hardware setups, model sizes, and parallelism strategies, characterizing the scaling properties of sharded training; training on up to 2048 H100 GPUs in Section 4.1 and 4.3 and studying models up to 70B parameters in Section 4.5

- Parallelization strategy recommendations which highlight that **model parallelism yields improved global throughput** despite prior work (Hagemann et al., 2023; Narayanan et al., 2021) and conventional knowledge suggesting that model parallelism lowers hardware utilization in Section 4.3.

- **Analysis of real-world cost metrics** showing that total GPU power draw and available FLOPS scale linearly with the number of devices, despite diminishing returns in throughput; resulting in reduced power efficiency and lower hardware utilization with greater parallelization (see Figure 1).

- **Comparisons across GPU hardware generations** suggesting that future improvements in computational throughput will only marginally improve overall throughput and power efficiency absent network fabric advancements and increased accelerator memory capacity in Section 4.4.

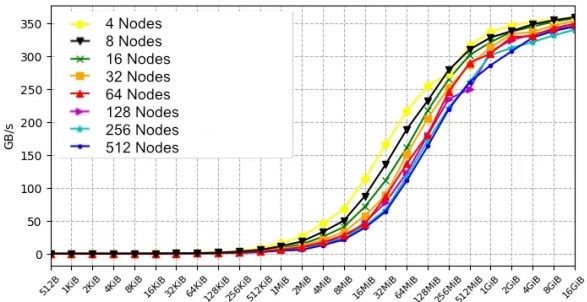 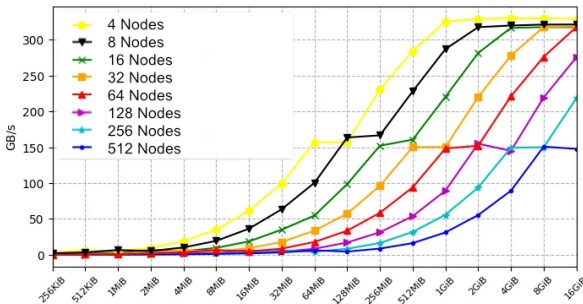

(a) Bandwidth of NCCL `AllReduce` using a tree algorithm and scales well with number of nodes (i.e. higher bandwidth).

(b) Bandwidth of NCCL `AllGather` using ring algorithms; scales poorly with the number of nodes (i.e. lower bandwidth).

Figure 2: Bandwidth measurements in GB per second of NCCL primitives on DGX H100 servers with eight GPUs per node, connected with InfiniBand, across world sizes from 4 to 512 nodes.

## 2 Preliminaries

In this section, we review commonly used parallelism techniques used in distributed training of large neural networks. The primary goals of distributed training are to: (1) enable model training with batch sizes and parameters that exceed the memory of individual GPUs; and (2) leverage the parallel processing power of additional hardware accelerators.

### 2.1 Parallelization Strategies

Below, we provide a brief taxonomy of commonly used distributed training algorithms and memory optimizations. In practice, these algorithms are not mutually exclusive and are often combined.

**Data parallelism** (Dean et al., 2012) replicates model parameters and optimizer states across GPUs with each device operating over a subset of examples in the global minibatch. After performing local forward and backward passes on their allocated minibatches, GPUs exchange and accumulate their partial gradients via an `AllReduce` collective such that each device obtains an identical global gradient and ensuring consistent model update. Data parallelism exhibits favorable communication properties as the `AllReduce` operation is non-blocking.

**Sharded Data Parallelism** alleviates the memory requirements of vanilla data parallelism by sharding model parameters, optimizer states, and gradients across a set of devices (referred to as a data parallel group); until weights are needed for computation or update. During computation for each layer, all parameters and optimizer states are obtained via `AllGather` on-the-fly such that at any time a GPU will maintain: the parameters and optimizer states for the current layer and its corresponding shard for all layers; `ReduceScatter` operations are subsequently used to update the weights and optimizer states during the backward pass. Each device performs all of the computation for each layer.

Fully-Sharded Data Parallelism Zhao et al. (2023) and DeepSpeed ZeRO Rasley et al. (2020); Rajbhandari et al. (2020) are commonly used sharded data parallelism strategies which enable training of large models without model parallelism. In contrast to standard distributed data parallel, sharded data parallelism introduces blocking communication operations to perform `AllGather` of model parameters; some of which can be overlapped by prefetch of subsequent layers during the previous layer computations.

**Model parallelism** shards model parameters across GPUs; each shard operates on the same minibatch simultaneously. In this setting, activations and their respective gradients are sent across GPUs.

- **Tensor Parallelism** (Shoeybi et al., 2019; Shazeer et al., 2018; Zheng et al., 2022) shards model parameters along hidden dimensions across a set of devices (referred to as a tensor parallel group) such that each GPU computes a partial sum of the intermediate activations, which are then aggregated across the tensor parallel group via an `AllReduce`. As the full set of activations are required for the subsequent layer, Tensor Parallelism introduces *blocking communication* for synchronization of intermediate activations across model parallel groups.

- **Pipeline Parallelism** (Huang et al., 2018; Harlap et al., 2018; Li & Hoefler, 2021; Li et al., 2021) shards model depthwise along with groups of layers being partitioned and allocated across devices; activations are then forwarded between devices via point-to-point communications. For all devices to be active at once, an input minibatch is

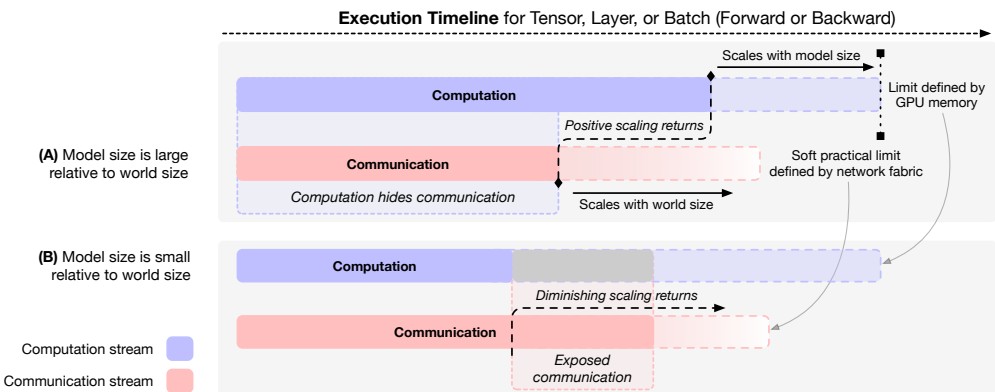

Figure 3: Two distinct training setups and their corresponding concurrent computation and communication streams, executing in parallel. In **(A)**, model computation size is large relative to world size; computation per-device hides communication cost and scaling the number of devices incurs minimal cost. In **(B)**, model computation size is small relative to world size. Communication is not hidden by computation and is exposed; scaling of world size incurs overhead and gives poor marginal gains in training throughput and efficient parallelization can be used to reduce communication overhead.

> split into microbatches which are staggered and pipelined according to various schedules (Narayanan et al., 2019; Lamy-Poirier, 2023). "Pipeline bubble" (Hennessy & Patterson, 2017), in which devices remain idle while awaiting data or instructions from other stages, reduces the efficiency of pipelining.

In contrast to sharded data parallelism, which reduces memory overhead by limiting the instantaneous parameters and weights on each device, model parallelism reduces memory overhead by limiting each device to perform computation on only a portion of the model. Sequence (Li et al., 2023; Korthikanti et al., 2023; Jacobs et al., 2023) and context parallelism (Liu et al., 2024; Yang et al., 2024) are techniques for reducing memory requirements for intermediate activations by partitioning and sharding along the sequence dimension. The joint combination of data, tensor, and pipeline parallelism techniques is frequently utilized in what is known as 3D parallelism to achieve higher communication efficiency (Shoeybi et al., 2019); or 4D parallelism when utilizing these methods along with context parallelism when training with longer sequence lengths (Dubey et al., 2024) .

**Communication-Computation Overlap**     Moving data over networks between accelerators utilizes distinct GPU resources unrelated to computation (e.g., dedicated copy engines, NVLink/NVSwitch) and can execute in parallel with computation. Overlapping communication and computation maximizes distributed training efficiency – it facilitates hiding communication latency, leading to near-perfect scaling. *Exposed communication*, that is communication which is executed without simultaneous computation leaves GPU's compute resources under-utilized.

## 2.2  Communication Primitives and Libraries

Modern deep learning frameworks (Paszke et al., 2019; Abadi et al., 2015; Bradbury et al., 2018) leverage specialized collective communications libraries, such as NCCL, RCCL, or XLA. In particular, we focus on NCCL which is used as the communication library for distributed operations across Nvidia GPUs as a representative hardware accelerator of data center training settings. In Figure 2, we empirically benchmark the `AllReduce` and `AllGather` operations performance with the NCCL library. The `AllReduce` collective is used in vanilla distributed data parallelism and tensor parallelism to aggregate parameter gradients and intermediate activations, respectively. `AllReduce` is supported by both Tree and Ring based algorithms in NCCL and observes favorable scaling properties as nodes increases. In contrast, `AllGather` and `ReduceScatter` are used for parameter rematerialization and gradient updates by FSDP and ZeRO and is supported by Ring algorithms in NCCL as of time of experimentation. `AllGather` and `ReduceScatter` and quickly becomes latency-bound as the number of devices increases. In general, the cost of communication collectives is expected to increase with the number of devices conducting operations. However, we note that differences in scaling patterns may exist when conducting communication on other network topologies (e.g. TPU pods) or with other communication libraries.

# 3 Experimental Methodology

In the following sections, we investigate the effects of scaling training workloads on end-to-end system performance and communication and computation volume. In particular, we conduct experiments across: distributed parallelization strategies, numbers of accelerators, hardware generation, model sizes, and input shapes (i.e. context length). Additional details on hardware and framework configurations are provided in Appendix B.

**Model Architectures**   We conduct our experiments with the Llama-2 decoder-only transformer (Dubey et al., 2024; Touvron et al., 2023) as a representative large language model. We utilize the AdamW optimizer (Loshchilov & Hutter, 2019; Kingma & Ba, 2015) and train on examples with a context length of 4096 and tokenized with a vocabulary of 32K; with data sampled from Wikipedia and StackExchange. Computation and most `AllGather` communication is performed in `bfloat16` precision; with reductions (`AllReduce` and `ReduceScatter`) performed in `float32` for numerical stability Liang et al. (2024); Rasley et al. (2020).

**Hardware Configuration**   We evaluate distributed training on datacenter clusters containing 8-GPU NVIDIA DGX nodes from the Ampere (80GB A100) and Hopper (80GB H100) architectures, with additional experiments on Volta GPUs (32 GB V100) in Appendix 15. We conduct our primary experiments on hardware scales between 1 and 32 eight-GPU nodes, with additional experiments up to 256 nodes, or 2,048 GPUs – to simulate pretraining scales.

**Parallelization Strategies**   We examine data, tensor, and pipeline parallelization strategies (colloquially known as 3D parallelism as described by Shoeybi et al. (2019); Rasley et al. (2020) and used in Dubey et al. (2024); BigScience Workshop (2022). Models are trained with Fully-Sharded Data Parallelism with explicit prefetching and without parameter resharding during the forward pass (i.e. FSDP, Zhao et al. (2023)) as in Llama-3.1 training equivalent to DeepSpeed ZeRO Stage 2.

We examine a range of group sizes for tensor and pipeline parallel strategies for, as described in Section 2, ranging from group sizes of 1 (i.e. single GPU training with no parallelization) up to group sizes of 16 (i.e. requiring parallelism groups across multiple nodes). Specifically, we analyze a range the parallel configurations resulting from the Cartesian product of tensor and pipeline parallelism sizes of {1, 2, 4, 8, 16} – parallelism configurations for all experiments are provided in Appendix C.

**Performance Metrics**   To understand the effects of both hardware and model scaling on end-to-end global and local per-device performance hardware utilization, we examine the following performance and efficiency indicators:

- **Throughput** is the rate at which examples are processed. We compute the estimated per-device *words per second* (WPS) and the global words per second across all devices.
- **Computational and communication load** is measured as the total execution time for CUDA and NCCL kernels, respectively. We calculate the total computation and communication load by aggregating CUDA and NCCL kernels from PyTorch execution traces.
- **Communication efficiency** is measured as the time in which communication kernels are exposed or overlapped with concurrent computation.
- **Hardware utilization** is measured as the number of floating point operations per second (FLOPS); alternatively, as Model FLOPS Utilization (MFU, Chowdhery et al. (2023)) which is the observed FLOPS as a percentage of the hardware's reported theoretical maximum.
- **Power utilization** is reported as the per-GPU power draw measured as the the average power draw with NVML

Metrics are aggregated from 60 training iterations, discarding the first 10 iterations to allow for stabilization of performance during the initial training iterations. Reported metrics are aggregated for the last 50 iterations.

# 4 Performance Analysis

## 4.1 Weak Scaling: Variable Global Batch Size

We first consider a *weak scaling* setting in which the *per-device workload* is kept constant as the number of GPU accelerators is increased. Each device carries a data parallel replica of a Llama 7B model with a local batch size of 2 examples, and is trained with FSDP without any model parallelism. This is representative of training settings in which there are insufficient devices to train a model without gradient accumulation; and increasing the number of devices can be used to reduce the number of gradient accumulation steps.

Figure 4: In FSDP training of Llama-7B, scaling the number of nodes and data parallel replicas *reduces hardware utilization and power efficiency* due to increasing exposed communication derived from increases in the size of communication kernels relative to fixed size computation kernels. Global throughput observes sub-linear scaling despite approximately linear increases in the total power utilization with number of nodes. "Ideal Hardware Scaling" corresponds to expected throughput assuming additional accelerators yield linear increases in throughput.

In Figure 4, we examine the effects of weak scaling of data parallel training instances across increasing numbers of accelerators from 8 GPUs up to 2048 GPUs. As expected, increasing the number of devices yields increases in global throughput as global batch size increases (i.e. Gustafson's Law for weak scaling; Gustafson (1988)). At small scales (i.e. when training using a limited number of devices), the cost of collective communication kernels is low relative to the cost of computation – and the communication overhead of weak scaling is minimal as non-blocking communication from FSDP can be hidden by executing data transfer and computation operations concurrently.

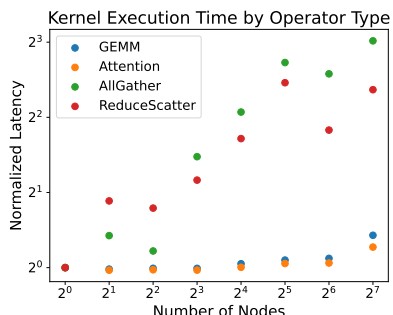

Figure 5: The relative execution time of both `AllGather` and `ReduceScatter` collectives scale with hardware world size.

However, as discussed in Section 2, increasing the degree of sharded data parallelism incurs larger collective communication costs for materialization of parameters via `AllGather` during the forward pass and gradient updates during the backward pass via `ReduceScatter`; with the latency of both operations scaling with number of nodes as observed in Figure 5. As a result, the total execution time for NCCL communication kernels and volume of exposed communication scales with the number of compute nodes limiting the extent to which weak scaling can be applied to sharded data parallel training – matching the expected behavior observed for the communication collectives in Figure 2b.

While the communication volume scales with node count, the per-device CUDA computation kernel execution time remains constant and becomes dominated by communication. As a result, exposed communication is unavoidable at scales *larger than 128 GPUs* and the hardware utilization decreases as there is insufficient computation to saturate the GPUs while waiting for the execution of larger communication kernels – this results in reductions the marginal speedup of global throughput and decreased local throughput as the number of devices increases.

While the per-device throughput scales sublinearly with the number of devices, the total power utilization scales approximately linearly which results in substantially worse real-world efficiency in GPU-hours and energy impact (i.e. fewer tokens processed per watt). When scaling from 128 to 2048 GPUs, the observed TFLOPS and words-per-second throughput decrease by 37.22% due to increasing exposed communication. Although the accelerator is largely idle on large scales and operates at lower arithmetic intensity, the power draw per GPU is roughly constant, only decreasing by 5.87% from 658W to 620W. As a result, the *overall power efficiency of the system likewise decreases with hardware scale* as seen in Figure 4.

## 4.2 Strong Scaling: Fixed Global Batch Size

We now examine the effects of strong scaling the number of accelerators to train workloads with a *fixed global batch size*, which results in decreasing effective local per-device batch sizes as the number of devices increases. This is representative of industry settings where excess compute resources can be allocated for a single training run; and there is a desire to minimize the time to complete a training run as opposed to maximizing the hardware utilization.

In Figure 6, we show that when training with a fixed global batch size of 32 examples across 2 to 32 nodes – allocation of additional devices yields diminishing returns in global throughput and reduced local hardware utilization and

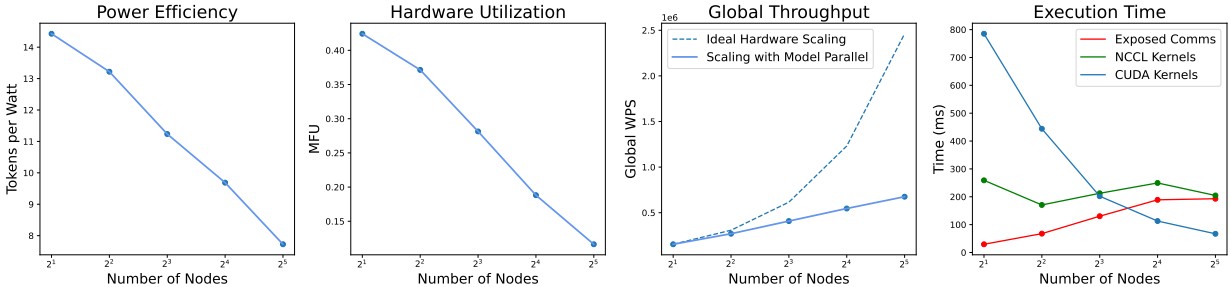

Figure 6: *Training with Fixed Global Batch Size Over Increasing Number of Nodes.* We utilize increasing degrees of model parallelism to distribute a fixed workload with global batch size of 32 across increasing numbers of GPUs. We select optimal model parallelism strategies according to the experimental results displayed in Figure 7. Even with optimal parallelization strategies, local throughput and hardware utilization declines with world size.

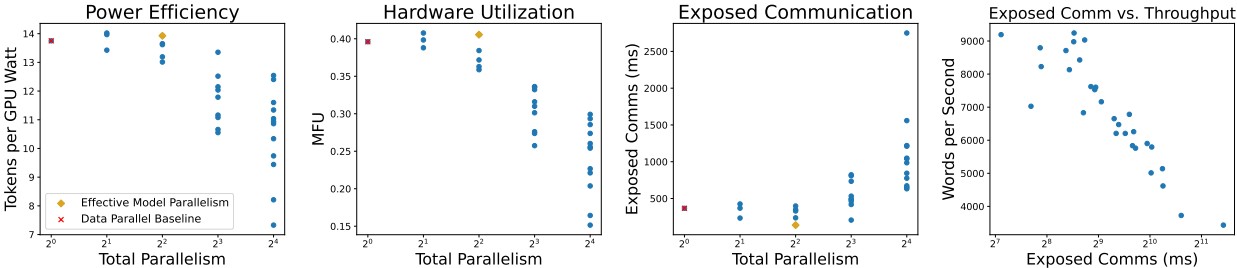

Figure 7: *Model parallelism increases FSDP throughput.* In model parallel training of Llama-7B with a fixed global batch size (512) and fixed number of accelerators (256 GPUs), there exist model parallel strategies that *increase* training throughput, hardware utilization, and power efficiency by reducing the total exposed communication (e.g. tensor parallelism of degree 4).

power efficiency. To distribute a fixed workload across more devices, it is necessary to introduce excess degrees of model parallelism which results in insufficient amounts of computation being allocated to each accelerator; which we observe as reduced execution time for CUDA kernels. At sufficiently large scales, excess parallelism causes previously compute-bound workloads to become communication bound and yields reductions in hardware utilization, which we observe in decreases in MFU from $40\%$ when training with 2 nodes to less than $15\%$ with 32 nodes. Practically, the overheads of strong scaling are especially apparent when using more than 4 nodes or 32 GPUs, as per-device workload sizes decrease and model parallelism becomes necessary.

In Appendix E, we conduct additional strong scaling experiments at full pretraining scale training both Llama-7B and 70B models on between 512 to 2048 GPUs, with limited marginal returns for increasing the number of hardware accelerators and observe decreases in MFU local hardware utilization by more than $30\%$.

## 4.3 Scaling Model Parallelism

As observed with both strong and weak scaling, fully sharded data parallel training of large neural networks suffers from communication bottlenecks when conducted over sufficiently parallel hardware platforms due to increasing costs of `AllGather` and `ReduceScatter` at scale.

Model parallelism is commonly used to complement data parallel training and reduce the memory requirements of a training workload to fit within the memory of each individual device. Additionally, model parallelism enjoys another beneficial property in which it can reduce the sizes of collective communication operations; as separate data parallel replicas are maintained for each model parallel group (i.e. data parallel collectives are executed over world sizes of $\frac{\text{Number of Devices}}{\text{Total Degree of Model Parallelism}}$, rather than over the Total Number of Devices) – where Total Degree of Model Parallelism is the product of Tensor and Pipeline parallelism group sizes.

In Figure 7, we search viable parallelism strategies for Llama 7B on 32 nodes with an effective local batch size of two and observe that small degrees of total model parallelism (i.e. tensor or pipeline parallel degrees of 2 or 4) reduce the amount of *exposed communication* and *increase throughput*. Although both tensor and pipeline parallelism introduce

additional communication operations, both techniques reduce the data parallel group sizes of the FSDP `AllGather` and `ReduceScatter` collectives; yielding higher throughput, hardware utilization, and power efficiency.

Furthermore, in Figure 8, we find that both tensor and pipeline parallelism are effective in reducing exposed communication; yielding higher words-per-second relative to the data parallel baseline. When scaling to more devices, we observe that the size of collective communications grows which necessitates increasing degrees of model parallelism to reduce the size of FSDP collective communications in Figure 11.

Notably, there is a limit to the extent to which model parallelism reduces exposed communication and improves throughput – as the `AllReduce` kernels required for Tensor Parallelism and bubbles introduced by pipeline parallelism grow with the degree of model parallelism. These communication costs become especially large when the parallelism occurs over multiple nodes as it relies on slower internode fabric (e.g. InfiniBand) – as noted in Figure 8, where there is substantial increases in exposed communication for tensor and pipeline parallelism strategies which are sharded at larger than 8 devices (i.e. across multiple nodes).

## 4.4 Scaling Hardware Speeds

In Figure 8, we examine the effects of scaling the hardware speed with comparisons between DGX-A100 and H100 clusters. In both cases, there exist model parallelism configurations which both increase the overall throughput and reduce the amount of exposed communication relative to data parallel baselines (i.e. total model parallelism of one).

When comparing the training performance of previous generation A100 to faster H100 hardware, with the optimal parallelization strategy for each platform, the MFU hardware utilization *decreases* from $59.67\%$ to $40.77\%$

The reduction in hardware utilization can be attributed to increases in exposed communication ($+12.83\%$) that emerge due to asymmetric improvements in communication and computation speeds (i.e. `bf16` `FLOPS` more than triples whereas NVLink and HBM bandwidth increase by ~$50\%$; See Table 1).

Between the A100 and H100 architectures, the extent to which training is *communication bound increases further with hardware generation*. Improvements to computation speed that outpace increases in data transfer speeds, result in computational kernels executing more quickly which make overlap with communication difficult (See Table 1). In Appendix H, we conduct additional experiments with V100 GPUs in which we confirm that the highest throughput is achieved with model parallelism.

## 4.5 Scaling Size of Model Architecture

We examine the effects of scaling the size of the neural network architectures across 1B, 7B, 13B, and 70B parameters. One might assume that increases in model parameterization solely increases the size of computation while leaving communication unaffected. However, as the number of parameters in a model scale, the volume of communication required for parameter materialization and gradient scattering increases jointly with the size of the computational operations (i.e. matrix operations with larger hidden dimensions). In Figure 9, we consider the optimal model parallelism strategy for each model architecture

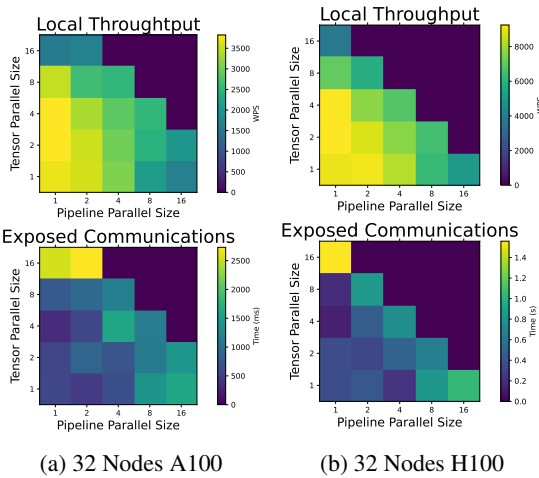

(a) 32 Nodes A100          (b) 32 Nodes H100

Figure 8: *Model Parallelism Improves Throughput.* Increasing degree of either tensor and pipeline parallelism yields improved throughput and less exposed communications compared to data parallel baselines.

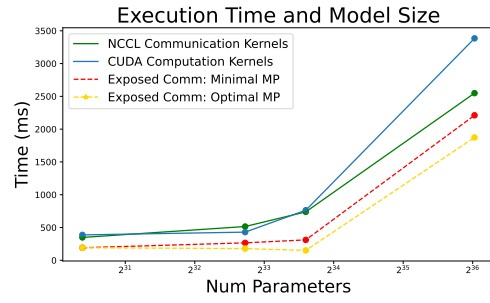

Figure 9: *Communication & Computation Both Scale with Model Size.* As computation load increases with model size, so does total and exposed communication. At all model scales, model parallelism reduces exposed communication.

by sweeping viable tensor and pipeline parallel configurations and observe that the volume of *exposed communication* likewise increases with model size, resulting in lower hardware utilization as models scale.

Additionally, we find that across architecture scales, there exist model parallelism strategies beyond the data parallel baseline or the minimal degree of model parallelism (for the 70B parameter model) that reduce the volume of exposed communication for all model sizes; and yield higher hardware utilization and throughput.

### 4.6 Scaling Context Length

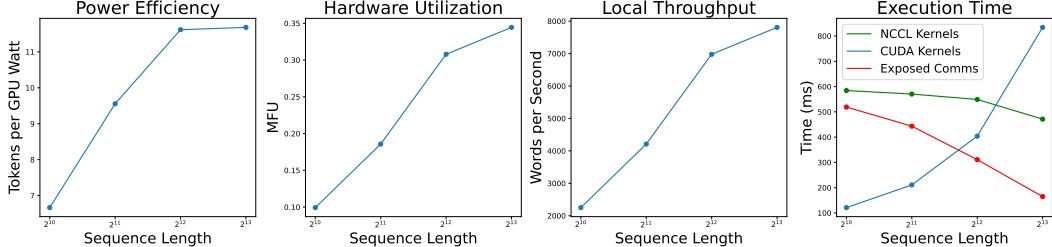

Figure 10: Increased sequence lengths yields larger compute kernels which better overlap with NCCL communication kernels, resulting in *lower exposed communication, higher hardware utilization and power efficiency*.

Finally, we examine the effects of varying the context length in Figure 10. When GPU memory is available, increasing the sequence length increases the computational workload allocated to each device without increasing the communication load, yielding improved the throughput, hardware utilization and power efficiency. However, reparameterization of the training process in this manner is often infeasible, as alterations to per-batch sequence length affects training dynamics predicted by computation-architecture scaling laws (Kaplan et al., 2020; Hoffmann et al., 2022).

## 5 Recommendations and Implications

We summarize our findings with directions for future work and best practices for researchers and practitioners.

**Model parallelism alleviates growing communication costs of FSDP.** Prior work (Hagemann et al., 2023; Shoeybi et al., 2019) studying 3D parallelism for large-scale training is conducted without the use of FSDP or ZeRO, and concludes that data parallel training is generally preferred to model parallelism when models fit within device memory. However, the collective communication primitives required by FSDP and ZeRO exhibit worse scaling properties than those used by standard distributed data parallel, as seen in Figures 2, 4, 6. We demonstrate that the increasing communication overhead of FSDP at scale can be mitigated by both tensor and pipeline parallelism.

In particular, we observe that standard FSDP training of 7B LLMs becomes unavoidably communication bound when training on more than 128 H100 GPUs. Beyond this scale, tensor parallelism at degrees of 2 or 4 achieves better or comparable throughput to the FSDP baseline. In our largest experiments at 2048 GPUs, introducing tensor parallelism yields a +52.60% increase in WPS throughput while only drawing 30W more in average GPU power per-device.

**Power efficiency and hardware utilization exhibit diminishing returns at scale.** As number of devices scales, energy efficiency decreases because the per-device computational throughput (FLOPS) decreases, despite power utilization remaining roughly constant (See Figure 4 and 6). Inefficient scaling and parallelization methods will worsen the energy efficiency and environmental cost of model training, as architectures and hardware platforms grow in scale (Strubell et al., 2019; Luccioni & Hernandez-Garcia, 2023; Luccioni et al., 2024; Schwartz et al., 2020). Rather than relying on synchronous training with a single large model, research in alternative training formulations that reduce communication overhead are required to improve model efficiency as models scale; such as via federated averaging, asynchronous training, ensembles and modular model architectures.

**Asymmetric improvements in hardware increase communication boundedness.** Hardware improvements have resulted in disproportionate growths in compute speeds that have outpaced improvements to memory and network speeds. As a result, model training is increasingly communication bound with the identical training workload observing a nearly 20% decrease in hardware utilization when using H100, as compared to A100 hardware (Section 4.4).

When training at large scales, faster interconnects are needed in addition to improvements in accelerator speed. Likewise increasing node size, such as with NVIDIA's GB-200 [1], connects more devices with high bandwidth memory and will allow for greater use of parallelism and alleviate communication boundedness.

**Performance measures and scaling laws must be compute and communication optimal**    Total number of Floating Point Operations (FLOPs) is commonly used to guide the development of efficient model architectures and compute-optimal scaling laws (Hoffmann et al., 2022; Tay et al., 2023; Dehghani et al., 2022). Without properly accounting for communication dynamics, performance measures and scaling laws cannot be extrapolated from small to large-scale. Integrating holistic information about hardware into scaling practice is essential given that collective communication dominates execution time at scale; scaling laws should be both *compute and communication optimal*.

# 6   Related Work

**Methods for Training at Scale**    While data, tensor and pipeline parallelization and FSDP are among the most common methods for distributed training of large neural networks, other approaches have been developed to the memory limitations and communication overhead of distributed training.

To address GPU memory limitations, numerous solutions have been proposed which: reduce the storage requirements of training workloads; or utilize offloading to lower bandwidth CPU memory. Activation checkpointing (Griewank & Walther, 2000; Chen et al., 2016) reduces peak memory utilization by discarding intermediate activations during the forward pass and recomputing activations for gradient calculation during the backward pass as needed. Strategies that determine optimal schedules for activation recomputation have been developed to manage the trade-off between activation memory and computational costs using hand-designed schedules or constraint solvers (Jain et al., 2020; Korthikanti et al., 2023; Yuan et al., 2024).

Alternatively, activation compression and reconstruction is an alternative approach to alleviate memory pressure to checkpointing (Evans & Aamodt, 2021; Georgiadis, 2019; Liu et al., 2021; 2022). Both approaches trade off additional computational overhead for reduced memory utilization. Heterogeneous CPU-GPU methods extend the memory sharding approaches introduced by FSDP and ZeRO to offload parameters, gradients, and optimizer states to larger RAM and NVMe memory (Rajbhandari et al., 2021; Ren et al., 2021). However, these methods incur substantial data transfer costs relying on CPU and PCI-E memory bandwidth orders of magnitude slower than GPU memory.

Communication overhead increases as the number of devices increases, which requires methods to reduce communication load. Hierarchical parallelization strategies such as Hybrid-Sharded Data Parallelism (HSDP, Ott et al.) Algorithmic variations of standard minibatch SGD reduce communication volume by performing less frequent parameter updates via federated averaging and asynchronous updating, such as Diloco, Local SGD, Model Soups, and Branch-Train-Merge (Douillard et al., 2023; Stich, 2018; Li et al., 2022; Wortsman et al., 2022). However, such methods exhibit distinct training dynamics from standard synchronous gradient-based methods (Charles et al., 2025). In contrast to the algorithmic and sample efficiency of training algorithms, we focus on efficiency via increased hardware and energy utilization.

**Evaluations of Parallelization Strategies.**    Previous studies empirically evaluating the scaling properties of distributed training strategies for neural networks has largely focused on the interaction of model parallelism with standard data parallelism techniques in the absence of FSDP or ZeRO-3 parallelism (Hagemann et al., 2023; Narayanan et al., 2021). Such studies recommend that total model parallelism be minimized due to the additional communication operations and overhead introduced by model parallelism, which we show does not apply when training with FSDP of Zero-3, alone.

Complementing empirical studies, automatic parallelization strategies and cost models for distributed training have been developed; such as Alpa, Galvatron, and FlexFlow (Zheng et al., 2022; Miao et al., 2022; Lu et al., 2017). However, these works limit their validation with smaller models and fewer accelerators (up to 64 GPUs) far less than the world sizes we evaluate in our experiments.

**Scaling Properties of Deep Learning.**    Previous work investigating the scaling properties of neural network training has largely studied the effects of varying the data volume, training compute budget, and model architecture (Hoffmann et al., 2022; Kaplan et al., 2020; Tay et al., 2023; Porian et al., 2024). These works primarily examine the impact of

---

[1]NVIDIA GB-200 Datasheet

these factors on the pretraining loss and downstream finetuning performance of the model with respect to the theoretical amount of computational resources allocated (i.e. number of FLOPs).

However, these analyses assume that workload performance scales directly with the amount of computation regardless of the underlying hardware platform and frameworks. In practice, theoretical measures (i.e. FLOPs) are known to be imprecise representations of end-to-end real-world performance (e.g. latency, throughput) due to performance bounds that emerge from management of the computational graph, data transfer, and communication bottlenecks (Dehghani et al., 2022; Fernandez et al., 2023) – or as we highlight due to communication boundedness.

## 7 Conclusion

In this work, we examine the effects of hardware scaling during the large-scale distributed training of large language models. Specifically, we conduct a comprehensive study of the impact of parallelization strategies, model architectures, and hardware platforms on throughput and energy efficiency during scaling with sharded data parallelism. We highlight that while sharded data parallelism is effective at reducing memory utilization when training in smaller regimes, communication boundedness dominates large-scale distributed training and results in reduced hardware utilization.

We show that communication boundedness worsens at scale and with newer hardware generations, and are persistent across model sizes. Additionally, we show that these trends lead to the emergence of viable model parallelism alternatives for distributing deep learning training workloads in contrast to existing recommendations and best practices in regards to training parallelization. Finally, we show that these trends culminate in significant diminishing returns on training performance with respect to real-world resources of power and throughput.

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

## A  Limitations and Statement of Broader Impact

In this work, we consider the set of data and model parallelization techniques for distributing training of neural networks – primarily focusing on the interactions of scale and parallelization in the Fully Sharded Data Parallel Setting as commonly used in the training of many open models. However, there are additional methods for workload parallelization and memory footprint reduction such as activation checkpointing, Hybrid Hierarchical Sharded Data Parallelism, 3D Parallelism without ZeRO or FSDP, and asynchronous algorithmic methods for optimization; which each utilize different computation and communication patterns and may exhibit differences in scaling behaviors.

In our investigation across computing platforms, we primarily consider variations in the speed of compute (i.e. GPU generation). In future work, we plan to demonstrate the consistency of the observed trends across settings with variable speeds of communication (i.e. varying speed of internode fabric by comparing InfiniBand interconnects with common alternatives such as RDMA over Converged Ethernet, RoCE).

Additionally, our work is focuses on the training of neural networks based on the Llama 2 transformer neural network architecture and GPU hardware accelerators. Although we expect our findings to be consistent across other model architectures, it is the case that even other transformer model architectures may vary in their choice of self-attention, position embeddings, tokenization, dropout, and norm layers which may affect communication and computation volume.

Likewise, we focus our investigations on GPUs as it is the most commonly used and easily available hardware accelerator. We expect that similar trends and tradeoffs between communication and computation would occur for alternative hardware accelerator architectures such as TPUs, IPUs, etc. However, other hardware platforms may exhibit differences in network topology and communication patterns which we reserve as settings for future study.

One of the primary goals of our work is to provide guidance and best practices for researchers and practitioners training large language models in order to reduce the computational, financial, and environmental impact of training. However, in doing so, this may incentivize further growth in training workloads leading to greater energy use and environmental harm from model training (i.e. Jevon's Paradox).

# B    Software, Hardware, and Dataset Details

Training is conducted in `bfloat16` precision with a Megatron-inspired framework and further optimizations provided by FlashAttention-2 (Dao, 2024) and xFormers (Lefaudeux et al., 2022). For our primary experiments, we trained models using PyTorch 2.3.1 built with CUDA 12.1, with attention implementation provided by XFormers 0.27. We utilize PyTorch FSDPv2 with prefetch of subsequent layers enabled.

For the A100 and H100 clusters, intra-node GPU communication occurs via fully connected second and third generation NVLink with NVSwitch, respectively. Inter-node communication occurs over an Infiniband fabric with 200 GB/s and 400 GB/s per-node bandwidth, respectively.

In supplementary experiments with V100 GPUs in Appendix H, models are trained in `fp16` with loss rescaling and CUTLASS (Thakkar et al., 2023) attention kernels on Volta hardware – due to limited hardware support on older Volta hardware. Nodes within the V100 cluster consist of 8-GPU setups connected with first-generation NVLink in a Hybrid Cube Mesh (HCM) topology.

We compute the runtime of communication and computation kernels by using PerfettoSQL to query Kineto profiles extracted by the PyTorch profiler, which is built on top of NVidia CUPTI to identify relevant NCCL and CUDA kernels, respectively. In Table 1, we provide additional details on the hardware platforms used for running our experiments.

The Llama 2 model is used via the Llama Community License and Acceptable Use Policy. Wikipedia and StackExchange data was made available via Creative Commons Attribution-ShareAlike 4.0 International License (CC BY-SA).

|                                          | V100 [2]     | A100 [3]     | H100 [4]     |
| ---------------------------------------- | ------------ | ------------ | ------------ |
| Tensor Core BF16 FLOPS                   | 125 TFLOPS   | 312 TFLOPS   | 990 TFLOPS   |
| GPU HBM                                  | 900 GB/s     | 2 TB/s       | 3.35 TB/s    |
| NVLink (GPU to GPU Comm)                 | 300 GB/s     | 600 GB/s     | 900 GB/s     |
| Internode InfiniBand (Node to Node)      | 100 GB/s     | 200 GB/s     | 400 GB/s     |

Table 1: Nvidia Reported DGX-Node Specifications by Generation.

# C    Parallelism Configuration Details

Below we provide the parallelism configurations swept for experiments in Section 4.

|                    | Global Batch Size          | Node Count           | Tensor Parallelism | Pipeline Parallelism | GPU Type            | Model Architecture    | Sequence Length         |
| ------------------ | -------------------------- | -------------------- | ------------------ | -------------------- | ------------------- | --------------------- | ----------------------- |
| Weak Scaling       | [16, 32, 64, 128, 256, 512] [5] | [1, 2, 4, 8, 16, 32] | 1                  | 1                    | H100                | 7B                    | 4096                    |
| Strong Scaling     | 32                         | [2, 4, 8, 16, 32]    | 1                  | 1                    | H100                | 7B                    | 4096                    |
| Model Parallelism  | 512                        | 32                   | [1, 2, 4, 8, 16]   | [1, 2, 4, 8, 16]     | H100                | 7B                    | 4096                    |
| Hardware Platform  | [256 [6], 512]             | 32                   | [1, 2, 4, 8, 16]   | [1, 2, 4, 8, 16]     | [V100, A100, H100]  | 7B                    | 4096                    |
| Model Size         | [256, 1024 [7]]            | 32                   | [1, 2, 4, 8, 16]   | [1, 2, 4, 8, 16]     | [1B, 7B, 13B, 70B]  | H100                  | 4096                    |
| Context Length     | [256, 512]                 | 32                   | 1                  | 1                    | H100                | 7B                    | [1024, 2048, 4096, 8192] |

Table 2: Parallelization Configurations Swept for Experimental Results.

---

[2]NVIDIA DGX-1 V100 Whitepaper

[3]NVIDIA DGX A100 Whitepaper

[4]NVIDIA DGX H100 Whitepaper

[5]For weak scaling experiments, models are trained with a fixed local batch size of 2, global batch size scales 1:1 with node count.

[6]Due to memory limitations, V100 experiments are only conducted with a local batch size of 1.

[7]Larger global batch size was used for the 1B parameter model to ensure higher GPU utilization.

# D   Additional Experiments: Model Parallelism in Alternate Settings

We extend the experiments from Section 4.3, in which we examine the effectiveness of model parallelism via Tensor and Pipeline parallelism with evaluations of other hardware settings and computational workloads. Here, we consider the effects of model parallelism in settings with lower hardware utilization, due to either: (1) smaller per-device workloads as determined by reduced effective local batch sizes (Figure 11a); or (2) larger communication loads from training in a increasingly distributed hardware settings (Figure 11b). In both regimes, there are a larger number of viable model parallelism strategies.

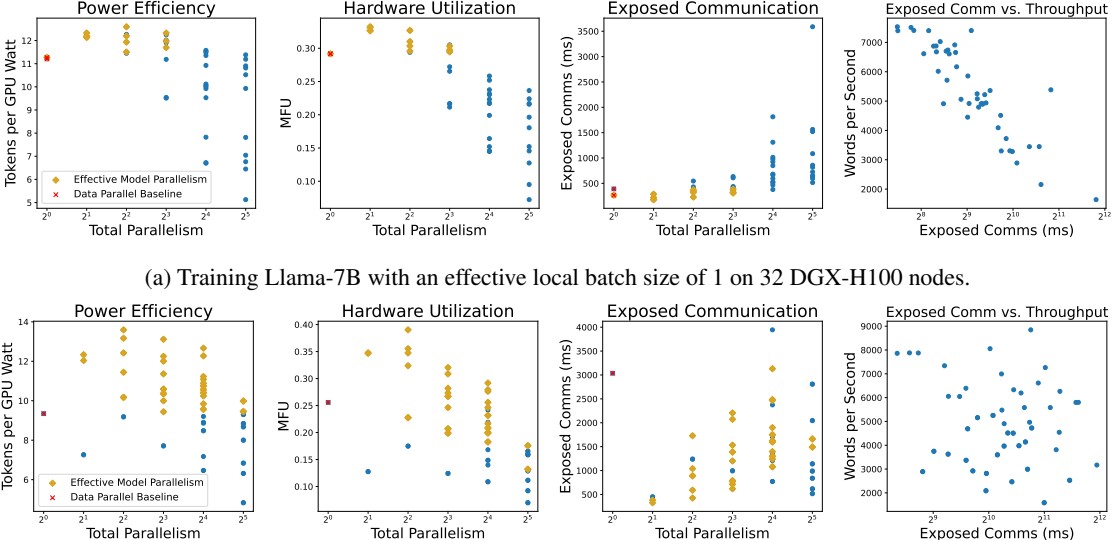

(a) Training Llama-7B with an effective local batch size of 1 on 32 DGX-H100 nodes.

(b) Training Llama-7B with an effective local batch size of 2 on 256 DGX-H100 nodes.

Figure 11: In regimes that are low in arithmetic intensity or communication bounded, there exist many viable strategies for model parallelism that: alleviate communication boundedness, increase power efficiency and hardware utilization.

# E Additional Experiments: Fixed Global Batch Size at Pretraining Scale

We extend the experiments from Section 4, in which we increase the allocation of hardware accelerators to a fixed computational workload with a constant global batch size – i.e. increasing the degree of parallelism across more accelerators without increasing the local effective batch size hardware utilization.

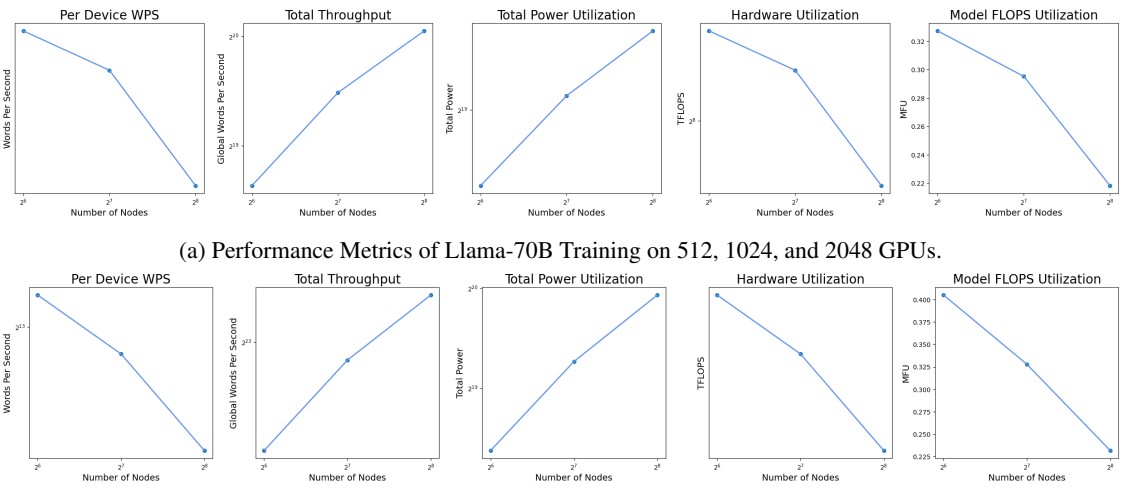

(a) Performance Metrics of Llama-70B Training on 512, 1024, and 2048 GPUs.

(b) Performance Metrics of Llama-7B Training on 512, 1024, and 2048 GPUs.

Figure 12: At pretraining scale, both Llama-7B and 70B observe regressions in hardware utilization and per-device local throughput as the number of devices is increased for a fixed computational workload.

# F Effects of Scaling on Memory Utilization

In fully-sharded data parallelism (FSDP), increasing the number of data parallel instances decreases per-GPU memory utilization by sharding parameters and gradients additional data parallel instances. However, memory savings diminish with device world size.

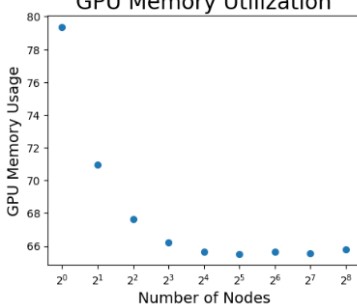

Figure 13: Increasing the data parallel group size reduces local per-GPU memory utilization, but reductions diminish with scale.

## G    Additional Experiments: Context Parallelism

We extend the results Section 4.3, to examine an additional form of parallelism, context parallelization Dubey et al. (2024). We use the context parallelization implementation provided by Nvidia's TransformerEngine. As context parallel is primarily used for very long contexts in Llama-3.1 with sequence lengths of 131,072, we find that context parallelism is a sub-optimal alternative to standard tensor parallelism for relatively common shorter sequence lengths of 4096.

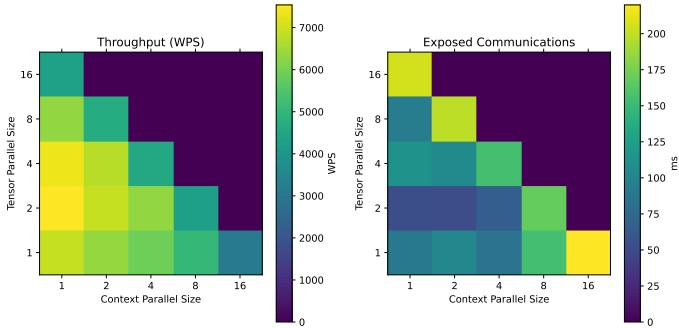

Figure 14: Effectiveness of Context Parallelism in training a Llama-7B model on 4k sequence length with H100 GPUs.

## H    Additional Experiments: V100 Hardware

In addition to our experiments in Section 4.3, we conduct additional experiments using older V100 GPUs from the Volta architecture training a Llama-7B model with an effective local batch size of 1 on 32 nodes. We observe similar trends in which small degrees of model parallelism improve overall throughput at scale. However, due to lack of optimized kernels (e.g. CUTLASS vs FlashAttention kernels) and Ampere hardware optimizations, we observe that the transition to Ampere A100 GPUs in fact improves overall hardware utilization.

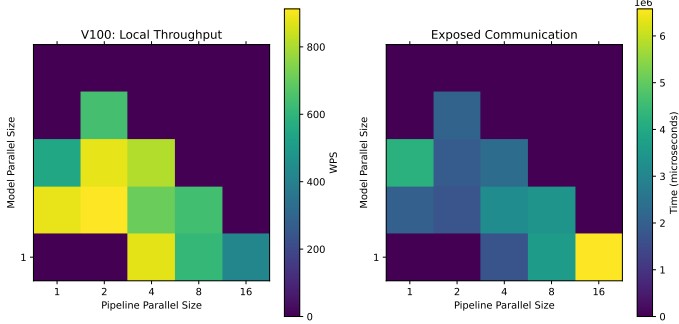

Figure 15: Throughput & Exposed Communication for Tensor Model Parallelization and Pipeline Parallelism Strategies on V100.

