# OpenReview forum: "Efficient Hardware Scaling and Diminishing Returns in Large-Scale Training of Language Models"
_TMLR — Accepted by TMLR_

### Review · Reviewer_iaff · 2025-04-25

**Summary Of Contributions:**

The backdrop of this paper is that modern large language models (LLMs) have many billions of parameters and their training is extremely compute intensive, which usually requires distribution across multiple GPUs and different forms of parallelism to perform efficiently. This paper presents an empirical study of distributed training across different setups. The authors explain what the different kinds of parallelism can be and analyze their performance. Crucially, they make recommendations on improving the training process in terms of real-world cost metrics and compare across different hardware generations.

Some of the key findings from the paper are:
- Overall power efficiency of the system decreases with hardware scale.
- Overheads of scaling become apparent when using more than 4 nodes or 32 GPUs.
- Small degrees of model parallelism reduce the amount of exposed communication and increase throughput.
- The extent to which training is communication bound increases with hardware generation.

The key overall contribution of the paper is that communication overheads worsen with increasing scale.

**Audience:**

Yes

**Broader Impact Concerns:**

No concerns on the Broader Impact.

I envision this paper as having significant positive impact to LLM practitioners.

**Claims And Evidence:**

Yes

**Requested Changes:**

Mitigating the two weaknesses outlined in the previous section would make for a better paper – i.e. experimentation on a GPT family model and a model with 100B+ parameters. In my opinion, these would strengthen the work, but are not critical to securing my recommendation for acceptance. The existing submission is already quite strong.

**Strengths And Weaknesses:**

Strengths:
- The paper is very timely, since LLMs are rising in prominence and getting larger, with their training costs increasing.
- The background on different types of parallelism - e.g. data, tensor, pipeline, context, etc - is clearly laid out and explained. This makes the paper excellent reading for practitioners of LLMs, who will benefit from the study and recommendations.
- The paper's novelty lies in the new experiments it performs to compute metrics. These are well-grounded in empirical evidence across different hardware and training configurations.

Weaknesses: The paper only considers Llama-2 models. While Llama-2 is a popularly used open source model, there are 2 issues with only considering Llama 2:
- The Llama family is different in internal structure from the GPT family of architectures, with some differences showing in token embeddings and dropout layers. While the core attention blocks largely remain the same, it would have been nice to see analysis on a representative model of the GPT family, e.g. GPT-NeoX.
- The paper considers LLMs up to 70B parameters, while the biggest open source models have > 100B parameters. This normally wouldn't have been important, but in a paper that specifically deals with considerations of distributed training, it would have been nice to see a 100B+ model.

---

> ### Author Response · Authors · 2025-05-11
>
> We thank the reviewer for their critique and feedback -- and are glad that they found this work useful in addressing the efficiency and training costs of LLMs.  We appreciate their comments regarding the comprehensiveness, potential positive impact, and utility of our work!
>
> We agree with the reviewer that further investigation of architectural variations can provide insight into finer-grained dynamics between computation and communication load. Unfortunately, at this time we are unable to conduct due to the scale of resources required for extension. However, we believe that some of their concerns may be addressed via the current set of experimentation.
>
> 1. **Model Scale:** Although our experiments are limited to models up to 70B, we believe the consistency of trends up to this size provide support for the presence and trends in FSDP communication overhead as model sizes grow. In particular, the 70B models are sufficiently large such that they require non-trivial degrees of model parallelism to fit in memory of 80GB GPUs; as required for training of larger LLMs.
>
> 2. **Other LLM Architecture Families:**  Although we are unable to experiment with GPT architectures, we note that the trends in communication overhead persist across all of the models examined. Within the Llama 2 family there exists architectural variations across parameter sizes. Specifically, 70B Llama 2 models utilize Grouped Query Attention as opposed to standard multi-head attention in the smaller models. For both the 70B and smaller models, we observe the same communication boundedness of FSDP at scale and improvements when using model parallelism.
>
> In an update to the Limitations section of Appendix A, we detail potential architectural variations which may be studied in future work.
>
> If the reviewer has any additional concerns, please let us know and we would be happy to address!

---

> > ### Comment · Reviewer_iaff · 2025-05-12
> >
> > I have viewed the updates to Appendix A and they look good to me.

---

### Review · Reviewer_5EwG · 2025-04-29

**Summary Of Contributions:**

This work empirically demonstrates that scaling LLM training using Fully Sharded Data Parallelism (FSDP) faces significant communication bottlenecks at large scales (hundreds to thousands of GPUs). These bottlenecks, primarily from AllGather/ReduceScatter operations inherent to FSDP, lead to diminishing returns in throughput and poor hardware utilization (MFU), even with newer, faster GPUs like the H100. The authors show that strategically introducing moderate degrees of model parallelism (tensor or pipeline) can effectively mitigate these FSDP communication overheads by reducing the size of communication groups, thereby improving overall training efficiency and throughput compared to using FSDP alone. Ultimately, the study highlights the critical need to consider communication costs alongside computation and advocates for communication-aware parallelization strategies and scaling laws for efficient large-scale training.

**Audience:**

Yes

**Broader Impact Concerns:**

I do not have any concerns regarding the ethical implications of this work.

**Claims And Evidence:**

Yes

**Requested Changes:**

1. Add a brief paragraph acknowledging whether any training stability issues were observed with the different parallelization strategies (especially those involving higher degrees of model parallelism) and confirm that the reported throughput metrics correspond to stable, converging runs. If specific measures were needed for stability, mention them.
2. Add a more explicit discussion in the main body (perhaps Section 5 or 6) acknowledging how the observed communication patterns (especially AllGather/ReduceScatter scaling) might differ significantly on other hardware, with different software stacks and different communication libraries.

**Strengths And Weaknesses:**

**Strengths**:
1. The work is grounded in a large-scale empirical study, utilizing up to 2048 H100 GPUs and models as large as 70 billion parameters over 50 iterations, which makes it quite relevant to industrial and research training settings.
2. By focusing on FSDP's interaction with model parallelism and hardware scaling, the study addresses practical challenges faced by engineers and researchers training modern LLMs, filling a gap left by previous studies that often didn't incorporate these details.
3. The work provides concrete, actionable recommendations, such as the counter-intuitive finding that incorporating model parallelism (tensor or pipeline) often improves throughput and efficiency when combined with FSDP at scale.

**Weaknesses**:
1. It doesn't investigate whether the different parallelization strategies (especially higher degrees of model parallelism) might affect model convergence dynamics or final downstream task performance for a fixed training duration or compute budget. It's possible that certain strategies could alter numerical precision or gradient dynamics in ways that impact training quality, not just speed.
2. The findings are specific to the PyTorch FSDP implementation, NCCL communication library, and NVIDIA's hardware stack. The conclusions might not directly translate to environments using different software or hardware stacks.

---

> ### Author Response · Authors · 2025-05-11
>
> We thank the reviewer for their feedback. We appreciate the reviewer’s recognition of the novelty and actionability of our empirical study of distributed training strategies!
>
> Below we address the reviewer’s concerns and requested changes.
>
> > Stability of Training with Model Parallelism:
>
> To verify the stability of our parallelization strategies on model training, we examined the training logs from our reported experiments. We confirm that all reported runs were stable over the 50 training iterations benchmarked (i.e. training loss decreased between the initial and final iterations).
>
> In regards to measures for numerical stability, communications for the reduction operations (i.e. `AllGather` and `ReduceScatter`) were conducted in `float32` precision  for numerical stability in all training runs while computation was conducted in `bfloat16` precision – as in TorchTitan and DeepSpeed frameworks [1,2]. We have updated Section 4 (Methodology) to reflect this detail.
>
> > Hardware and Framework-Specific Results.
>
> As the reviewer notes, our study studies GPUs as a representative accelerator of deep learning training hardware as they constitute the majority of accelerators used in distributed training.
>
> In an update to Section 3 (Preliminaries), we include a revised statement acknowledging that our work focuses on the scaling of collective communications on GPUs; and note the accelerators may utilize alternative network topologies (e.g. TPU 3D pods) which may exhibit different communication scaling properties as world size increases.
>
> Please let us know if there are any other concerns that we can address!
>
> References
> 1.  Liang, Wanchao, et al. "TorchTitan: One-stop PyTorch native solution for production ready LLM pre-training." arXiv preprint arXiv:2410.06511 (2024).
> 2. Rasley, Jeff, et al. "Deepspeed: System optimizations enable training deep learning models with over 100 billion parameters." Proceedings of the 26th ACM SIGKDD international conference on knowledge discovery & data mining. 2020.

---

### Review · Reviewer_PwSH · 2025-04-30

**Summary Of Contributions:**

This manuscript provides a thorough end-to-end empirical analysis of how fully sharded data parallelism (FSDP) scales in practice. Profiling Llama style architectures across parameter counts, parallelism configurations, sequence length, and accelerator hardware versions on up to 2048 NVIDIA GPUs, the authors empirically show that NCCL AllGather and ReduceScatter traffic becomes the dominant cost beyond 128 GPUs when using FSDP. To alleviate this issue, the authors demonstrate that adding a modest amount of model parallelism (PP/TP) degree shrinks those collectives and recovers a significant amount of efficiency at large cluster scales with only a small power increase. Experiments across V100, A100, and H100 hardware show that faster GPUs alone cannot overcome the network bottleneck, motivating the paper’s practical guidelines for hybrid parallel layouts.

**Audience:**

Yes

**Broader Impact Concerns:**

Currently, there are no broader impact concerns.

**Claims And Evidence:**

Yes

**Requested Changes:**

1. (If possible) Analyses on hierarchical FSDP and 3D parallelism would benefit the paper.
2. To the best of my knowledge, the exact hybrid parallel configurations used in the experiments are not disclosed. It would benefit the paper if these configurations were provided in the Appendix.
3. The legend of Figure 5.c says model parallel, but as far as I know, the figure is talking about data parallel?

**Strengths And Weaknesses:**

**Strengths**:

1. A key strength of this work is its empirical, large-scale evaluation of FSDP under realistic training workloads. The experiments are conducted on up to 2,048 GPUs, providing real world experimental data on large clusters for FSDP scaling across various parameter counts, parallelism configurations, sequence lengths, and even hardware versions, which are hard to obtain. Thus, the experiments offer valuable insights into communication bottlenecks that emerge at such a scale.
2. The paper also offers practical guidance on hybrid parallel strategies that improve throughput and efficiency in large clusters.

**Weaknesses**:

1. The paper lacks comparison with Hierarchical FSDP (e.g., FSDP grouping for 32 nodes, DDP across each 32-node group).
2. The paper lacks explicit comparison with 3D parallelism, which is also a very widely accepted method (a preliminary comparison could provide many insights).

---

> ### Author Response · Authors · 2025-05-11
>
> We thank the reviewer for their feedback and appreciate their recognition of the scale and insightfulness of our experimentation!
>
> Below we address the reviewer’s requested changes.
>
> > Analysis of 3D and HSDP Parallelism.
>
> We agree that Hierarchical FSDP and 3D parallelism are relevant methods for comparison,  and may exhibit differences in communication patterns and scaling behaviors.  Unfortunately, the training framework utilized for our experimentation did not have support for HSDP or vanilla distributed data parallelism for 3D parallelism (i.e. without sharding) at the time of experimentation. In Appendix A, we add a note on the limitations of our focus on FSDP and that these other methods may exhibit differing scaling patterns.
>
> With our current results, we believe that our existing work provides substantial support for the diminishing returns in scaling FSDP. Such results provide utility for researchers and practitioners where FSDP is commonly used as a standalone approach for scaling distributed model training (e.g. OLMo, Mosaic MPT, OpenELM, etc).
>
> > Specification of Hybrid Parallelization Configurations.
>
> In revision to Section 3: Experimental Methodology, we provide explicit specifications of the parallelism configurations swept.
>
> Our primary experiments with hybrid data-model parallelism configurations consist of a sweep over the cartesian product of tensor and pipeline parallel sizes of [1,2,4,8,16]  x  [1,2,4,8,16]; with data parallel size determined by the local or global batch size. To reflect these details, we have updated Appendix C with a table specifying the parallelism configurations used in each of the experiments in Section 4: Results, as requested by the reviewer.
>
> > Legend on Figure 5c.
>
> In Figure 5, we examine the performance of strong scaling with *fixed global batch size*. In this setting, degree of model parallelism increases with the number of GPUs, as additional sharding is needed to distribute the fixed workload over more devices. As a result, increasing the number of devices does not scale the number of data parallel replicas.
>
> We have added additional details to the caption of the corresponding Figure to clarify why model parallelism is the target axis for scaling.
>
> Please let us know if there are any concerns we can address!

---

> > ### Comment · Reviewer_PwSH · 2025-05-13
> >
> > Thank you for the response and manuscript revisions.
> >
> > Regarding discussion point #1, while I think comparison with alternative parallelism strategies would be insightful, I also acknowledge that it could be cumbersome and costly to conduct such comparisons. Since the authors revised the manuscript to mention other strategies but limit the scope of discourse to FSDP only, I think it is now fairly self-contained, so I think the concern is addressed.
> >
> > For discussion point #2, a minor and optional revision for Figure 7 could be: to include the exact configuration for the "Effective Model Parallelism" point in the caption.

---

### Review · Reviewer_Qhdk · 2025-05-02

**Summary Of Contributions:**

This work empirically explore the diminishing hardware efficiency of LLM training under popular paralleization strategies when scaling the number of GPUs.  The work conducts a comprehensive scaling study on hundreds-to-thousands of GPUs for LLM training that also examines different parallelization mixes and measures energy use.  If I understand correctly, this fills a gap between theoretical scaling law papers and low-level systems studies by providing an integrated view of model, software, and hardware scaling. Overall, the messages are useful for both researchers interested in the limits of scaling and practitioners planning large training runs. The insight that *the choice of parallelization strategy should change at an extreme scale* (due to FSDP’s communication overhead) is intriguing and meaningful.

 The experimental methodology is thorough and generally sound.  I particularly like the discussion on two settings: weak scaling setting (increasing global bs) and strong scaling (fixed global scaling). This discussion is useful and it shows the rigor of the authors.

The paper is well-written and the results are presented with clarity.  I post some minor presentation suggestions as follows to enhance the readability for a broader range of readers. **I am not sure if the challenge of diminishing returns in distributed computing is well-known in the high-performance computing community,  but the message is relevant to a broad range of researchers in LLMs.**

**Audience:**

Yes

**Claims And Evidence:**

Yes

**Requested Changes:**

**Some questions:**

1.  Based on your results, small degrees of model parallelism can significantly improve throughput once communication becomes a bottleneck (e.g., beyond 128 GPUs for a 7B model). How should a practitioner determine the *optimal degree of tensor/pipeline parallelism* for a given model and hardware setup? Any concrete suggestions?

2. I think we need more explanations in Section 2 to distinguish SDP and Model parallel (MP).

    In section 2, the authors define SDP as "Sharded Data Parallelism alleviates the memory ..  by sharding model parameters .. in a data parallel-group;" Meanwhile, the authors also  define model parallel by " Model parallelism shards model parameters across GPUs; each shard operates on the same minibatch simultaneously." Without sufficient background, **it is unclear how SDP and MP are different from each other.** From my understanding  (correct me if wrong): SDP would “shard the matrix but still run the whole matrix multiplication on every GPU"; while  MP “shard matrix and do partial matrix multiplications on single”. As a result, FSDP  requires Allgather but MP does not. Is this understanding correct? Please provide more explanations and backgrounds.

3. In SDP, what does it mean by "in a data-parallel group"? Does "one data parallel group" mean "one GPU"?

4. The paper focuses on hardware efficiency, which is orthogonal to algorimthic convergence speed (or training speed). It would be great if the authors make a disclaimer in the early sections of the paper.

**Strengths And Weaknesses:**

See above

---

> ### Author Response · Authors · 2025-05-11
>
> We thank the reviewer for their feedback and are glad they found our work intriguing and insightful!
>
> Below we address the reviewer’s questions, and include references to manuscript updates for their requested changes.
>
> > **Identification of Optimal Parallelism Degrees:** How should a practitioner determine the optimal degree of tensor/pipeline parallelism for a given model and hardware setup? Any concrete suggestions?
>
> For practitioners, we suggest sampling an execution trace of a training iteration to determine if the execution of AllGather and ReduceScatter communication operations are exposed, which can be determined via examination of the communication and computation streams. Optimal parallelism can be determined by incrementally increasing model parallelism until communication operations are overlaid with computation. We include a new conceptual model in Figure 3 to indicate when scaling and parallelism are useful.
>
> > **Explanations of Parallelization Strategies**: Sharded Data Parallelism vs Model Parallelism.
>
> The reviewer’s understanding of SDP and Tensor Parallelism are correct. In contrast to `AllGather` and `ReduceScatter` operations used by SDP to communicate model parameters, tensor parallelism requires `AllReduce` operations to communicate activations .  We have updated Section 2: Preliminaries to clarify the destination between how SDP and model parallelism reduce memory load.
>
> In sharded data parallelism, the model parameters and corresponding optimizer states are divided across a set of devices (the data parallel group), to reduce memory overhead until weights are needed for computation or update. During computation for a specific layer, all parameters and optimizer states for a specific layer are `AllGathered` onto each device on-demand such that at any time a GPU will maintain: all of the parameters and optimizer states for the current layer and its corresponding shard for all layers. Each device performs all of the computation for each layer.
>
> By contrast, in tensor parallelism, each device shards parameters column-wise for all layers across a set of devices (the tensor parallel groups). Each device performs only a component of the computations for each layer using the full set of activations and the shard of weights. The final layer output is obtained by aggregating the partial sum activations across the tensor parallel group via an `AllReduce` to obtain a full set of activations on each device.
>
> > Defining Data Parallel and Tensor Parallel Groups
>
> Data parallel groups correspond to the number of devices over which weights and optimizer states are sharded; equivalently in vanilla FSDP it corresponds to the total number of model replicas.  We have specified this in Section 2.
>
> > Hardware vs Algorithmic Efficiency.
>
> As the reviewer notes, the focus of this work is on improving hardware and network utilization for distributed training as opposed to sample efficiency and model convergence. We have included a statement reflecting this distinction in the Related Work section of the updated manuscript.
>
> Please let us know if there are any other concerns that we can address!

---

### Decision · Action_Editor_JsQS · 2025-05-27

**Recommendation:** Accept as is

**Comment:**

All of the reviewer provides positive comments for the paper. The core contribution lies in the observation about parallel training, which is insightful for the ML community.

**Audience:**

The paper discusses an interesting problem that would be interesting for the machine learning community.

**Claims And Evidence:**

The paper is technically solid, and the claims and statements in the paper are introduced with concrete evidence.